# A Multi-Mediation Analysis of the Association between Adverse Childhood Experiences and Non-Suicidal Self-Injury among South African Adolescents

**DOI:** 10.3390/ijerph21091221

**Published:** 2024-09-17

**Authors:** Steven J. Collings, Sachet R. Valjee

**Affiliations:** School of Applied Human Sciences, University of KwaZulu-Natal, Mazisi Kunene Road, Glenwood, Durban 4041, South Africa; valjees@ukzn.ac.za

**Keywords:** adverse childhood experiences, non-suicidal self-injury, emotion dysregulation, adolescents, South Africa

## Abstract

The available research findings suggest that non-suicidal self-injury (NSSI) constitutes an important public health issue, with identified risk factors for NSSI having been found to include exposure to adverse childhood experiences (ACEs) and comorbidity with various mental disorders. However, the available findings have, for the most part, been based on the experiences of individuals living in predominantly high-income countries located in the Global North, and it is not clear whether these findings can be confidently generalised to individuals living in low-resourced countries. As such, this cross-sectional study assessed risk factors for NSSI in a non-clinical sample of 636 South African adolescents (12–18 years old), with the data being analysed using a multi-mediation analysis. ACEs were assessed using a revised version of the ACEs Questionnaire, and NSSI was assessed using items adapted from the Self-Harm subscale of the Risk-Taking and Self-Harm Inventory for Adolescents, with emotion dysregulation, depression, and PTSD being considered as possible mediators. High prevalence rates for NSSI and exposure to five or more ACEs were reported by the participants, with the mediation analysis indicating that significant direct effects of adverse childhood experiences on NSSI were partially mediated by emotion dysregulation. These findings are discussed with respect to their implications for primary, secondary, and tertiary prevention.

## 1. Introduction

Non-suicidal self-injury (NSSI) has been included in the fifth revised edition of the *Diagnostic and Statistical Manual of Mental Disorders* as a condition that may be the focus of clinical attention (DSM-5-TR) [1], with NSSI being defined as intentional, self-initiated damage to the surface of the body (e.g., cutting, hitting, skin abrasion) in the absence of suicidal intent. Global estimates of prevalence rates for NSSI range from 4% to 23% for adults, with comparative rates for non-clinical samples of adolescents being somewhat higher at 13.9% to 28.6% [2,3]. It is generally acknowledged that NSSI constitutes a serious clinical and public health problem, as NSSI has been found to be associated with a variety of mental health problems—including depression, anxiety, borderline personality disorder, and eating disorders [4,5,6,7,8]—and to constitute a risk factor for suicidal behaviour [9,10,11]. As such, research on risk factors for NSSI would appear to be strongly indicated, with the available literature suggesting that risk factors for NSSI stem from exposure to various forms of childhood trauma, comorbidity of NSSI with many other disorders, and the various functions of NSSI [2].

It is also important to note that estimates of the prevalence and dynamics of NSSI have been found to vary across countries and continents [3]. However, such variations have been reported almost exclusively in relation to countries located in the Global North, with two recent meta-analyses of NSSI prevalence rates [3,12] including only two studies conducted in the Global South (one in Brazil and one in Australia), but no studies conducted in the African context. As such, there would appear to be a clear need for global estimates of NSSI prevalence rates and dynamics to be expanded to reflect a more truly global perspective, with such expansion appearing to be most strongly indicated in relation to the African context.

### 1.1. Childhood Adversity as a Risk Factor for NSSI

Adverse childhood experiences (ACEs) include a range of potentially traumatic experiences, occurring before the age of 18, that have been found to be associated with a cumulative risk for compromised physical, behavioural, mental, and social health [13,14]. Although exposure to ACEs has consistently been found to constitute a risk factor for NSSI [2,15,16,17,18,19,20], available studies have tended to construe ACEs somewhat narrowly, with the assessment of ACEs having largely been restricted to direct acts of child maltreatment occurring in the home (i.e., physical, emotional, sexual abuse, and/or neglect) and to adverse family circumstances (parental divorce or separation, mother treated violently, family member incarcerated, substance abuse in the home, and mental illness or suicide attempts in the home). As a result, we have little understanding of the association between NSSI and (a) children’s indirect exposure to ACEs in the home (e.g., witnessing domestic violence), which has been found to be associated with compromised physical and mental health outcomes [21], or (b) children’s exposure to potentially traumatic extrafamilial ACEs (e.g., exposure to community violence or peer victimisation), which has been found to be associated with deleterious physical and mental health outcomes [22].

Although child and adolescent exposure to discrimination (racial, sexual orientation, sexual identity) has seldom been conceptualised as an ACE, available evidence would suggest that discrimination has a stronger effect on mental health and executive functioning outcomes than conventional ACEs [23,24], with findings from studies conducted in high-income countries suggesting that discrimination is strongly associated with NSSI [25,26,27,28]. Further, despite the democratic elections of 1994, racial and other forms of discrimination continue to constitute a major public health concern in South Africa [29], with there being an associated need to explore the heuristic value of including discrimination as an additional ACE when assessing risk factors for NSSI in both low-to-middle-income and high-income countries.

### 1.2. Mechanisms Underlying the Association between ACEs and NSSI

There is an emerging body of research that suggests that there are two related, although somewhat distinct, pathways between ACEs and NSSI. First, from a neurobiological perspective, there is substantial support for the view that repeated exposure to ACEs is linked to changes in brain functioning and structure and in stress-response neurobiological systems, with these changes having been found to be associated with enduring effects on an individual’s physical and mental wellbeing as well as with deficits in emotional regulation (ED) and executive functioning [30,31,32,33,34]. Taken together, this perspective would predict a direct effect of ACEs on the development of mental disorders (such as NSSI) as well as a direct effect of ACEs on ED and impaired executive functioning.

A second and related pathway between ACEs and NSSI is suggested by the psychological theories of Chapman et al. and Nock [35,36], which attempt to account for why and how distress and ER associated with ACEs give rise to NSSI behaviours. According to these theories, the primary function of NSSI is to address uncomfortable emotional feelings (including ED) associated with a number of mental disorders, including depression and PTSD [37], with NSSI having been found to constitute an effective and efficient method of regulating affective and cognitive experiences associated with childhood trauma [38]. These psychological theories would predict that the association between ACEs and NSSI is likely to be mediated by feelings of distress and/or ED.

Empirical support for both the neurobiological and psychological pathways to NSSI has been reported in a number of studies. With respect to the neurobiological model, the available studies have consistently reported a direct effect of ACEs on NSSI [2,14,15,17,18,19,39,40] and a direct effect of ACEs on ED and/or emotional distress associated with PTSD and depression [2,41]. Similarly, with respect to the psychological model, studies have found that emotional distress and ED partially mediate the association between ACEs and NSSI [39].

### 1.3. Research on Factors Mediating the Association between ACEs and NSSI

We were able to identify seven studies that have examined mediators of the ACEs-NSSI association [19,38,39,40,42,43,44], with the potential mediators considered in these studies being consistent with the predictions suggested by contemporary conceptualisations of NSSI (i.e., ED, depression, and symptoms of posttraumatic stress disorder). Six studies considered ED as a potential mediator of the ACE-NSSI association, with ED being found to constitute a significant mediator or moderator of NSSI severity in five studies [38,39,42,43,44] but not in the study conducted by Shenk and associates [40]. The mediating effects of depression on NSSI severity were examined in five studies, with depression being found to mediate NSSI outcomes in four studies [19,38,39,42], but not in one study [40]. Finally, only one study [41] assessed the mediating effects of PTSD, with PTSD symptoms emerging as a significant mediator of NSSI severity, but with depression and ED not significantly mediating NSSI severity after controlling for PTSD.

The disparities in the available studies may reflect differences in study design. With respect to sampling strategies, study participants were drawn from the general community in five studies [19,38,39,42,44], with the mediating effects on NSSI outcome being assessed in a clinical sample of adolescents in one study [43] and a sample drawn from Child Protection Services referrals in another study [40]. Further, there was a lack of consistency regarding the role played by key variables in relation to NSSI outcomes. Although ED and depression were generally conceptualised as constituting mediating variables, ED was entered as a moderating variable in one study [38], with depression being entered in another study as a control variable [42].

In interpreting the available mediational findings, it is also important to bear in mind that there were marked differences across studies in the way that NSSI was operationalised, with some studies employing validated research instruments in order to obtain an estimate of NSSI severity [19,38,39,42,43] and two studies [40,44] relying exclusively on one or two broad questions to assess for NSSI (e.g., “Have you ever tried to intentionally hurt yourself by damaging the physical integrity of your body?”). Clearly, such broad questions fail to adequately address the nature and scope of NSSI as defined by the American Psychiatric Association [1].

At a broader level, the available mediation studies have been conducted exclusively in two continents—North America [38,40,43] and Asia [19,39,42,44]—and among participants residing in middle-income countries [19,39,44] or in high-income countries [38,40,42,43], with none of the identified mediation studies having been conducted in countries located in the Global South. To the extent that estimates of the prevalence and dynamics of NSSI have been found to vary across countries and continents [3], it is possible that the disparities in mediation findings may also reflect geographic and/or income level variations across study samples.

In sum, the available mediation studies are characterised by marked heterogeneity in sampling and design, with inter-study variability serving to limit the confidence with which cross-study comparisons can meaningfully be made.

### 1.4. The Present Study

The broad aim of the present research was to conduct a study on the association between ACEs and NSSI in a non-clinical sample of adolescents living in a low-resourced African country and to assess the extent to which the study findings correspond to trends that have emerged in the extant literature. In addition to assessing comparative prevalence rates for ACEs and NSSI, this study was designed to explore potential mediators of the ACEs-NSSI association, with the selection of potential mediators in this study being informed by both conceptual and empirical considerations. As we have indicated above, the contemporary conceptualisations of risk factors for NSSI are united in regard to ED as well as general psychological distress (including symptoms of depression and PTSD) as being important mediators of NSSI outcomes [27,28,29,30,31,32,33,34], with empirical support for the mediating role of ED, depression, and PTSD being suggested by findings from the limited number of studies that have been conducted on the topic [19,38,39,40,42,43,44].

As such, in this study, it was predicted that the impact of risk factors for NSSI outcomes would be mediated by ED, depressive symptoms, and symptoms of PTSD, with the conceptual model employed in the study being presented in Figure 1.

## 2. Materials and Methods

### 2.1. Participants

The convenience sample used in this study included all 734 students attending a South African secondary school. Six hundred and thirty-six participants (86.6%) completed usable questionnaires, with sample attrition being due to an absence of written consent/assent (*n* = 57, 7.8%), extensive missing data (*n* = 25, 3.4%), and participant voluntary withdrawal from the research during questionnaire completion (*n* = 16, 2.2%). An examination of school records indicated that participants did not differ significantly from non-participants in terms of age, ethnicity, and gender. Sample demographics were female (34.4%), age (12–18 years, *M*_age_ = 15.4, *SD* = 1.5), and ethnicity (96.0% Black African), with participant numbers being more or less equivalent across grade levels.

### 2.2. Measures

ACEs were assessed using a revised version of the ACEs Questionnaire (ACES-R) [22], which contains 5 items related to child maltreatment, 5 items related to adverse family circumstances, and 4 items related to broader contextual issues (low socioeconomic status, peer victimisation, social isolation, and witnessing community violence). Although the ACES-R has not previously been used with South African samples, the scale has been found to be associated with enhanced predictive validity in a large and representative sample of adolescents in the United States [22]. For the purposes of this study, an additional ACE (discrimination) was added to the ACE-R, with discrimination being operationalised using an item adapted from the Daily Discrimination Scale (DDS) [45]: “Before your 18th birthday, were you often or very often called names, insulted, or threatened because of your race, skin colour, sexual orientation, or sexual identity?” (0 = No, 1 = Yes to any form of discrimination). Given that ACE questionnaires are primarily designed to measure the confluence of multiple types of ACEs [21], ACEs in this study were assessed using a cumulative measure of ACEs (i.e., the total number of ACEs reported by participants), with potential scores ranging from 0 to 15.

NSSI was assessed using an adapted version of the Self-Harm (SH) subscale of the Risk-Taking and Self-Harm Inventory for Adolescents (RTSHIA) [46], which has previously been used in studies of South African adolescents [47]. In Vrouva and colleagues’ validation study [46], the SH subscale was found to have a high level of internal validity (α = 0.93) and acceptable levels of 3-month test–retest reliability (r = 0.87), as well as acceptable levels of convergent, concurrent, and divergent validity (α = 0.86 in the present study). The SH subscale contains eight statements related to intentional NSSIs (e.g., cutting, burning, biting, skin abrasion), with each statement beginning with the phrase “have you ever intentionally…”, with frequencies being rated on a 4-point scale ranging from never to many times. Consistent with recommendations that NSSI behaviours are likely to be most clinically meaningful if they occurred in the past year [6], each item on the SH subscale was adapted to assess past-year NSSI behaviours. For each item on the SH subscale, participants were asked to indicate the number of days that they had engaged in such behaviour in the past 12 months. Each item on this adapted version of the RTSHIA was scored using a 4-point scale (never to many times in the past year), providing a score range of 0-24.

ED was assessed using the 6-item Alteration in Regulation and Affect (ARA) subscale of the self-report version of the Structed Interview for Disorders of Extreme Stress Scale (SIDES-SR) [48], which has previously been used in research on South African adolescents [49]. Research on the ARA indicates that the scale has high levels of internal consistency (α = 0.82), with Cronbach alpha in this study being 0.80. The ARA provides a symptom severity score for ED ranging from 0 to 24.

Symptoms of depression were assessed using a nine-item subscale of the Patient Health Questionnaire (PHQ-9), which is used to assess for the severity of depressive symptoms [50]. The PHQ-9, which has been validated in samples of South African adolescents [51], has been widely validated, with Cronbach alpha across three trials ranging from 0.86 to 0.89 [50] and alpha in this study being 0.86. Each of the nine items on the PHQ-9 questionnaire is scored using a 4-point Likert scale that assesses how often individuals have been bothered by depressive symptoms in the past two weeks, with response options ranging from 0 (not at all) to 3 (nearly every day), providing an estimate of symptom severity ranging from 0 to 27.

The severity of PTSD symptoms was assessed using the 20-item Posttraumatic Stress Disorder (PTSD) Checklist for DSM-V (PCL-5) [52], which has previously been used in research on young adults in South Africa [53]. Blevins and colleagues report high levels of convergent and discriminative validity for the scale as well as strong internal consistency (α = 0.94) [48]. In this study, Cronbach alpha was 0.92. Items on the PCL-5 are scored on a 5-point scale ranging from 0 (not at all) to 4 (extremely), providing a score range of 0–80, with a score of 33 or more representing clinically significant levels of PTSD symptoms.

### 2.3. Procedure

This study was conducted in an urban co-educational public high school in the Durban Metropolitan Region (South Africa). The school selected for the research was chosen not simply for convenience purposes but largely because school staff had identified a specific problem area (learner behavioural and psychological problems) that they felt could, at least to some extent, be addressed by our research. Prior to questionnaire administration, a meeting was held with the school principal and staff to outline the nature and purpose of the research. School staff were enthusiastic about the research and agreed to arrange a special school assembly to inform students regarding what their participation would involve, with the voluntary nature of participation being emphasized. Students were then provided with informed consent forms to be completed by caretakers, with those who obtained caretaker consent being invited to complete an assent form. In cases where caretaker consent and child assent were provided, students were invited to complete a research questionnaire. Questionnaires were administered to whole classes of students during Life Orientation classes, with the researchers and Life Orientation teacher being present to assist with any queries participants may have had and to monitor participants for possible distress. In cases where distress was noted, participants were provided with the option to discontinue their participation.

### 2.4. Ethical Considerations

An institutional ethical clearance certificate was obtained for the research (protocol reference number: HSS/1029/013D). In addition, written gatekeeper permission from the school principal, parental consent, and participant assent were obtained, with continuous assent being employed to ensure that students could discontinue their participation at any time and with offers of free supportive counselling from the school counsellor being made to all participants.

### 2.5. Statistical Analysis

The data analysis involved three stages. Initially, data were reviewed, with this review revealing that the proportion of missing values was less than 3% across all study variables, which can be considered negligible as it falls well below the 5% rule of thumb for acceptable levels of missing data [54]. As such, for each analysis, listwise deletion was used to handle missing values. The data review also revealed that there were no outliers.

Second, prevalence rates were calculated for ACEs and NSSI, with descriptive statistics (means, standard deviations, and intercorrelations) being calculated for all key variables. Finally, a multi-mediation analysis was conducted using the PROCESS macro for SPSS [55]. After controlling for the effects of demographic variables (age, gender, and ethnicity), (a) the risk factor entered in this analysis was a cumulative measure of ACE exposure (i.e., the number of different types of ACE reported by each participant); (b) the outcome variable was the severity of NSSI; (c) the mediating variables were the severity of ED, depressive, and PTSD symptoms; and (d) bootstrapped confidence intervals (CIs), based on 5000 bootstrapped samples, were used to determine the significance of indirect effects.

## 3. Results

### 3.1. Descriptive Statistics

Of the 636 participants included in this study, 211 (33.2%, SE = 0.02) reported that they had engaged in NSSI behaviours on at least five days in the past year. The three most frequently reported forms of NSSI were “cut your skin” (23.3%, SE = 0.04), “banged your head against a hard surface” (19.7%, SE = 0.04), and “burned yourself with a hot object” (9.1%, SE = 0.02). The prevalence rates for NSSI were higher among 12–14-year-old participants (42.2%, SE = 0.04) than among 15–18-year-olds (29.1%, SE = 0.04; χ_2_^1*df*^ = 10.66, *p* = 0.001), with participant sex and ethnicity being unrelated to NSSI prevalence rates.

A total of 593 participants (93.2%) reported lifetime exposure to at least one ACE, with 187 (29.4%) reporting exposure to five or more ACEs. The four types of ACEs that were most strongly associated with NSSI severity were discrimination (*b* = 0.75, SE = 0.2), domestic physical abuse (*b* = 0.69, SE = 0.02), peer victimisation (*b* = 0.67, SE = 0.02), and sexual abuse in the home (*b* = 0.53, SE = 0.12). Additional descriptive statistics for continuous variables are presented in Table 1.

### 3.2. Mediation Analysis

The mediation model was tested, and the results are presented in Figure 2. ACEs were positively and significantly associated with ED (*b* = 0.23, SE = 0.03, *p* < 0.001), depression (*b* = 0.21, SE = 0.03, *p* =< 0.001), PTSD (*b* = 0.30, SE = 0.23, *p* < 0.001), and NSSI (*b* = 0.28, SE 0.02, *p* < 0.001). In addition, NSSI was positively and significantly associated with ED (*b* = 0.29, SE = 0.08, *p* < 0.001) and depression (*b* = 0.09, SE = 0.07, *p* = 0.033), but not with PTSD (*b* = 0.01, SE = 0.01, *p* = 0.898). The mediation analysis revealed a positive and significant indirect effect of ACEs on NSSI through ED (effect = 0.29, CI: 0.0149, 0.0484), but no significant indirect effect for depression (effect = 0.15, CI = −0.0649, 0.0288) or PTSD (effect = −0.041, CI = −0.0012, 0.0152). The direct effect of ACEs on NSSI in the presence of all mediators was also found to be significant (*b* = 0.015, SE = 0.03, *p* < 0.001). Taken together, these findings indicate that ED partially mediated the relationship between ACEs and NSSI.

## 4. Discussion

The primary aim of this research was to explore the mediating role of emotion regulation, depression, and PTSD on the association between ACEs and NSSI in a non-clinical sample of South African adolescents and to assess the extent to which the study findings correspond to the findings that have been reported in studies conducted largely in the Global North (mainly in Asia and North America).

With respect to prevalence rates, the present findings suggest that both ACEs and NSSI are highly prevalent among South African adolescents. Of the 636 participants in this study, 593 (93.2%) reported exposure to at least one ACE, with 187 (29.4%) reporting exposure to five or more ACEs. Although these prevalence rates are similar to the rates reported in previous studies of low-to-middle-income countries [56,57], they are markedly higher than the comparative rates reported for high-income countries [58,59]. Regarding the prevalence of NSSI, one in three participants (33.2%) reported that they engaged in NSSI on at least five occasions in the past year. These prevalence rates are notably higher than the 12-month global prevalence rate of 23% for repetitive NSSIs reported for non-clinical samples of adolescents [3]. Taken together, these high prevalence rates for both ACEs and NSSI suggest that adolescents living in Africa (and in other low-to-middle-income countries) face a comparatively higher risk of exposure to both ACEs and NSSI, with there being an associated need for primary, secondary, and tertiary intervention programmes designed to effectively address both the prevalence and consequences of both ACEs and NSSI in low-resourced countries.

This study’s findings indicate that the association between ACEs and NSSI is partially mediated by ED, with higher levels of ED being associated with increased NSSI severity. These findings are consistent with the neurobiological theories that posit that ACEs will be associated with the severity of both ED and NSSI [32] and consistent with the psychological theories that predict that the association between ACEs and NSSI will be mediated by emotion regulation [36]. As such, this study’s findings suggest that both theoretical perspectives may need to be considered when considering pathways to NSSI.

The finding that ED partially mediated the ACEs-NSSI association is congruent with findings from previous studies conducted in both middle- and high-income countries that have identified ED as one of the strongest mediators of the ACE-NSSI association [38,39,42,43,44]. As such, comprehensive NSSI interventions, designed to address both the direct effects of ACEs and the indirect effects of ED on NSSI outcomes, would appear to be indicated. In this regard, multi-phase interventions (such as trauma-focused cognitive behavioural therapy [60]) initially focusing on safety and addressing ED before going on to address traumatic events have been found to be more effective than single-phase interventions (which focus exclusively on addressing traumatic events), with this enhanced efficacy having been found to be most marked among children and youth [61].

The fact that there was a significant direct association between ACEs and NSSI in the presence of selected mediators suggests that there may be additional mediators that could usefully be explored in future studies, with the available studies suggesting a number of additional variables that could usefully be considered, including female gender [18], LGBTQI+ status [28], dissociation [34], and self-compassion [19]. Further, research on potentially traumatic life events that are prolonged and/or repetitive in nature has identified a trio of disorders of self-organisation (DOS) that are associated with chronic exposure to ACEs, with DOSs comprising (a) severe and prolonged problems with affect regulation, (b) persistent beliefs about the self as being diminished or worthless, and (c) difficulties in sustaining relationships or feeling close to others [53]. While the present study provides support for the mediating effects of affect dysregulation, further research could usefully explore the mediating effects of all three DOSs.

## 5. Conclusions

### 5.1. Study Findings

To a large extent, the findings relating to the prevalence and dynamics of ACEs and NSSI were consistent with the findings reported in the extant literature. For example, (a) the high prevalence rates for ACEs and NSSI observed in this study are consistent with the view that adolescents living in low-resourced countries face a higher risk of exposure to both ACEs and NSSI [62,63], and (b) the finding that ED most strongly mediated the ACE-NSSI association is consistent with findings from previous studies conducted in both medium- and high-income countries [19,38,39,42,43,44] that have identified ED as a significant mediator of NSSI risk. However, the fact that depression and PTSD were not found to mediate NSSI severity after controlling for ED was not anticipated, with this finding appearing to be worthy of further study.

### 5.2. Implications for Theory and Prevention

This study’s findings would appear to have implications for both theory and intervention. With regard to theory, this study’s findings are consistent with Kira’s development-based trauma framework [23,24], in terms of which trauma severity is evaluated in terms of three trauma types: Type I (single traumatic exposure), Type II (a sequence of events that occurred and then stopped), and Type III (continuous traumatic exposure that can continue throughout life), with there being compelling evidence that Type III traumas have the most severe traumagenic potential [23]. As such, it is hardly surprising that in this study *discrimination* (a Type III trauma) was more strongly associated with NSSI severity than predominantly Type I and II traumas that are assessed in standard ACE measures. It is similarly not surprising that ED, a condition that has been found to be strongly associated with chronic/continuous types of traumatic exposure [62], emerged as a significant mediator of NSSI severity in this study. The fact that Type III traumas have been found to be highly prevalent in low-resourced countries such as South Africa [64], with a history of discrimination having been found to be associated with NSSI in studies conducted in low-, middle-, and high-income countries [65,66,67], suggests that discrimination is an ACE that may have global relevance.

The fact that both ACEs and NSSI were found to be highly prevalent in this sample suggests the need for primary prevention efforts designed to reduce levels of ACE exposure (e.g., cash and kind transfers or home visitation programs) and/or to foster resilience among children, with available evidence suggesting that efforts to bolster children’s resilience need to be targeted at individual factors such as self-esteem and self-compassion [19,68] as well as at efforts to increase social capital at a domestic, peer, and community level [69,70,71]. With respect to secondary and tertiary prevention, there are a variety of interventions that have been specifically designed to address the direct and/or indirect effects of cumulative exposure to ACEs, with such interventions including emotion regulation therapy [72] and trauma-focused cognitive behavioural therapy [56]. Similarly, there are a number of psychotherapeutic interventions that have been found to be effective in treating NSSI among adolescents, with the most effective interventions in terms of reducing NSSI behaviours being interventions that use strategies from (or adapted from) cognitive behavioural therapy, Dialectical Behaviour Therapy, and/or Psychoeducation [73].

### 5.3. Strengths and Limitations of the Study

To our knowledge, this is the first study of factors mediating the association between ACEs and NSSI to be conducted on the African continent, with its findings suggesting that both ACEs and NSSI are highly prevalent in low-resourced settings and that ED is an important mediator of the ACE-NSSI association. The limitations of our study include the fact that data were obtained using a cross-sectional design that precludes strong causal inferences and the use of a questionnaire that relied on retrospective recall in relation to key constructs that may have been associated with recall bias. As such, future research on the association between ACEs and NSSI needs to ideally rely on prospective methods of data collection that are designed to address these limitations. In addition, the use of self-report measures of participants’ wellbeing falls short of the gold standard for assessing psychological wellbeing (i.e., a comprehensive clinical assessment). As such, future studies on the association between ACEs and NSSI would benefit from a reliance on a more comprehensive assessment of key constructs. Finally, male and Black African participants were over-represented in the study sample, which could potentially limit the generalisability of our findings. As such, further research based on more demographically representative samples is indicated.

## Figures and Tables

**Figure 1 ijerph-21-01221-f001:**
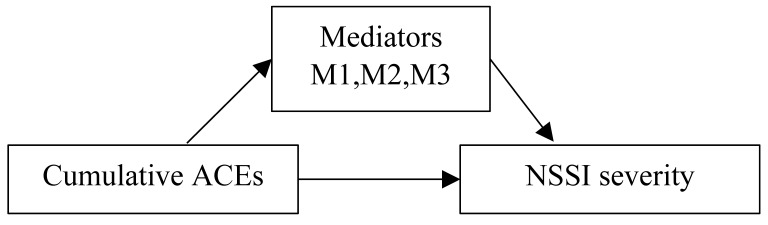
Conceptual model informing the research.

**Figure 2 ijerph-21-01221-f002:**
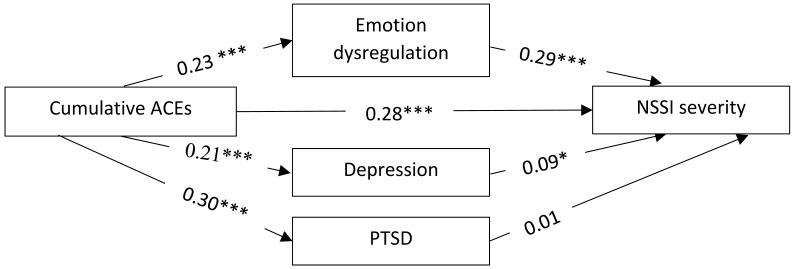
The results of the mediation analysis. The pathways between variables are indicated by standardised beta estimates. * *p* < 0.05, *** *p* < 0.001.

**Table 1 ijerph-21-01221-t001:** Means, standard deviations, and correlations of the variables included in the analysis.

			Zero-Order Correlations			
Variable	M	SD	1	2	3	4
1 ACEs	3.80	2.48	-			
2 ED	6.24	2.16	0.221	-		
3 Depression	9.01	3.18	0.191	0.224	-	
4 PTSD	17.19	12.27	0.292	0.187	0.214	-
5 NSSI	6.06	10.47	0.266	2.25	0.213	0.302

All coefficients are significant < 0.001 (two-tailed). ACEs = adverse childhood experiences; NSSI = non-suicidal self-injury; ED = emotion dysregulation.

## Data Availability

School authorities did not provide consent for us to share the data with any third parties.

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
