# Peer review of "A Multi-Mediation Analysis of the Association between Adverse Childhood Experiences and Non-Suicidal Self-Injury among South African Adolescents"

_ijerph, 2024, doi:10.3390/ijerph21091221_

Round 1

Reviewer 1 Report

Comments and Suggestions for Authors

The paper reports an examination of factors that may mediate the association between the experience of adverse childhood experiences (ACEs) and the engagement during high school years with non-suicidal self-injury. A novel feature of the study was the addition of a ‘discrimination’ item to an otherwise standardised questionnaire about adverse childhood experiences. Data were obtained via a cross-sectional survey of students from a single co-educational high school in South Africa (96% Black, two thirds male). This is the only study of this nature I have identified from South Africa. The field is dominated by papers from China and Taiwan. The authors chose to examine depressive symptoms, emotional dysregulation and post traumatic stress symptoms as putative mediators. Only emotional dysregulation played a significant mediating role between ACEs and NSSI.

Of specific types of ACEs, experience of discrimination had the strongest association with NSSI but as it was not among the commonly reported ACEs the relevance is uncertain.

The authors concluded that their findings are consistent with Kira’s development based trauma framework. They recommend South Africa invests in strategies to reduce ACEs such as a home visitation program.  There should also be strategies to increase children’s resilience. Finally, emotion regulation therapy and trauma-focuses CBT should be offered to children exposed to ACEs as a secondary prevention measure.

Comments to authors:

1.       Adverse childhood experiences needs to be included in the title as the study is concerned with the factors mediating the association between ACEs and NSSI

2.       The background to the study requires a more sophisticated review of the extensive and somewhat confusing literature concerning risk, mediating and moderating factors for NSSI in adolescents. Note that across different studies the same variable may be treated as a risk , mediating or moderating factor.

3.       I identified two further studies in addition to Hu et al  Shenk et al that have specifically examined the relative mediating effects of depression end emotion dysregulation on the relationship between ACEs and NSSI. They are Wu et al Personality and individual differences, 2021-06, Vol.175, p.1107 and Gong et al Journal of interpersonal violence, 2024-03, Vol.39 (5-6), p.925-948

4.       If no hypotheses were entertained how was it the authors landed on ACEs as the risk factor and depressive symptoms, emotion dysregulation and post traumatic stress symptoms as the mediating factors? Other putative risk factors for NSSI examined in research include:

a.       Screen time Wiguna et al Frontiers in psychiatry, 2021-11, Vol.12, p.743329-743329

b.       Problematic internet gaming Xu et al Comprehensive psychiatry, 2023-08, Vol.125, p.152398-15239

c.       Self criticism Gong et al Journal of adolescence (London, England.), 2019-04, Vol.72 (1), p.1-9

d.       Maladaptive perfectionism Gu et al Journal of clinical psychology, 2022-06, Vol.78 (6), p.1137-1150

e.       Parental expressed emotion Ammerman and Brown
Current psychology (New Brunswick, N.J.), 2018-03, Vol.37 (1), p.325-333

f.        Alexithymia Tang et al Child and adolescent psychiatry and mental health, 2022-06, Vol.16 (1), p.43-43

g.       Connection to parents and friends Taliaferro et al Clinical child psychology and psychiatry, 2020-04, Vol.25 (2), p.359-371

h.       Parental psychological control Huang et al Children and youth services review, 2022-05, Vol.136, p.106417, 

i.         Psychological need satisfaction Huang et al Children and youth services review, 2022-05, Vol.136, p.106417, 

j.         Emotion regulation Lan et al Psychology research and behavior management, 2022-06, Vol.15, p.1451-1463

k.       Self compassion Liu et al Psychological trauma, 2023-10, Vol.15 (7), p.1203-1213

l.         Early maladaptive schemas Babaeifrd et al BJPsych open, 2024-05, Vol.10 (3), p.e116-e116,

m.     Mindfulness Zhang et al Aggressive behavior, 2024-01, Vol.50 (1), p.e22121

n.       Parental emotional warmth Qin and Gan Children and youth services review, 2024-06, Vol.161, p.107662,

o.       Nightmares Ennis et al Comprehensive psychiatry, 2017-07, Vol.76, p.104-112

p.       Low perceived social support Mendez et al Journal of affective disorders, 2022-04, Vol.302, p.204-213

5.       Stronger justification for choice of mediating variables is required, particularly as it does not align with the cited systematic review of Valencia-Aguda et al Journal of adolescence (London, England.), 2018-06, Vol.65 (1), p.25-38. Note also the long list of putative mediating variables reported in papers published since the systematic review

a.       Anxiety Can et al Journal of affective disorders, 2024-02, Vol.347, p.57-65, Hou et al BMC psychiatry, 2023-04, Vol.23 (1), p.248-248, Long et al BMC psychiatry, 2024-01, Vol.24 (1), p.25-25

b.    Impulsivity Can et al Journal of affective disorders, 2024-02, Vol.347, p.57-6, Li et al Frontiers in psychiatry, 2023-05, Vol.14, p.1139705-1139705

c.    Parental support Liu et al BMC public health, 2024-07, Vol.24 (1), p.1-11

d.    Peer support Liu et al BMC public health, 2024-07, Vol.24 (1), p.1-11

e.    “psychological sub-health’ Huang et al Frontiers in psychiatry, 2022-02, Vol.13, p.798369-798369

f.     Alexithymia Ke et al Journal of affective disorders, 2024-04, Vol.350, p.295-303, Guo et al Psychology research and behavior management, 2024-01, Vol.17, p.783-797

g.    School engagement Yu et al  Frontiers in psychology, 2020-11, Vol.11, p.572521-572521

h.    Sleep disturbance Wang et al Frontiers in psychology, 2024-03, Vol.15, p.1325436-1325436, Rong et al Child abuse & neglect, 2024-03, Vol.149, p.106627-106627, Hou et al BMC psychiatry, 2023-04, Vol.23 (1), p.248-248

i.      Psychotic like events Wang et al Frontiers in psychology, 2024-03, Vol.15, p.1325436-1325436

j.      Rumination Wang et al Current psychology (New Brunswick, N.J.), 2024-06, Ying et al Child abuse & neglect, 2021-05, Vol.115, p.104992-104992

k.    Identity confusion Gu et al Child abuse & neglect, 2020-08, Vol.106, p.104474-8

l.      Maladaptive perfectionism Ying et al Child abuse & neglect, 2021-05, Vol.115, p.104992-104992

m.   Coping strategy Ren et al Journal of clinical psychology, 2018-07, Vol.74 (7), p.1246-1257

n.    Psychological resilience He at al Children and youth services review, 2022-02, Vol.133, p.106335

o.    Loneliness He at al Children and youth services review, 2022-02, Vol.133, p.106335

p.    Self-criticism Glassman et al Behaviour research and therapy, 2007-10, Vol.45 (10), p.2483-2490

q.       Mobile phone addiction Long et al BMC psychiatry, 2024-01, Vol.24 (1), p.25-25

r.     Sex differences Long et al BMC psychiatry, 2024-01, Vol.24 (1), p.25-25

s.    Self esteem Guo et al Psychology research and behavior management, 2024-01, Vol.17, p.783-797

t.     Alienation Gu et al Journal of interpersonal violence, 2023-02, Vol.38 (3-4), p.3864-3882

u.    Shame Keng et al International journal of forensic mental health, 2019-10, Vol.18 (4), p.293-304

6.       How does the school population studied (predominantly black, majority male) influence the generalisability of the findings? What other confounders could be contributing to learner behavioural and psychological problems?

7.       A novel aspect of the study was the inclusion of discrimination as an ACE item. The authors highlighted the importance of distinguishing types of ACE in NSSI research. I am puzzled therefore why different types of ACE was not included in the modelling?

8.       The conclusions are generic and do not relate specifically to the study findings. With respect to secondary and tertiary prevention the authors need to place their recommendations within the context of what is currently known about treatment for NSSI.

Author Response

Manuscript: A multi mediation analysis of the association between adverse childhood experiences and non-suicidal self-injury among South African adolescents

RESPONSE TO REVIEWER 1 COMMENTS

Comment 1: Adverse childhood experiences needs to be included in the title as the study is concerned with factors mediating the association between ACEs and NSSI.

Response 1: Thank you for pointing this out. We agree and have changed the title to: A multi mediation analysis of the association between adverse childhood experiences and non-suicidal self-injury among South African adolescents

Comment 2: The background to the study requires a more sophisticated review of the extensive and somewhat confusing literature concerning risk, mediating and moderating factors for NSSI in adolescents. Note that across different studies the same variable may be treated as a risk , mediating or moderating factor.

Response 2: Thank you for this comment. We agree and have added comments under subheading 1.3 in the introduction that specifically address the issue you raise plus other methodological issues relating to research on mediators of the  ACEs-NSSI association:

              Disparities in available studies may reflect differences in study design. With respect to sampling strategies, study participants were drawn from the general community in five studies [19,38,39,42,44], with the mediating effects on NSSI outcome being assessed in a clinical sample of adolescents in one study [43] and a sample drawn from Child Protection Services referrals in another study [40]. Further, there was a lack of consistency regarding the role played by key variables in relation to NSSI outcomes. Although ED and depression were generally conceptualized as constituting mediating variables, ED was entered as a moderating variable in one study [38], with depression being entered in another study as a control variable [42].

In interpreting available mediational findings it is also important to bear in mind that there were marked differences across studies in the way that NSSI was operationalised, with some studies employing validated research instruments in order to obtain  an estimate of NSSI severity (19,38,39,42,43], and two studies [40,44] relying exclusively on one or two broad questions to asses for NSSI (e.g., “Have you ever tried to intentionally hurt yourself by damaging the physical integrity of your body”). Clearly such broad questions fail to adequately address the nature and scope of NSSI as defined by the American Psychiatric Association [1].

At a broader level, available mediation studies have been conducted exclusively in two continents – North America [38,40,43] and Asia [19, 39,42,44] – and have been conducted among participants residing in middle income countries [19,39,44] or in high income-countries [38,40,42,43]; with none of the identified mediation studies having been conducted in countries located in the Global South. To the extent that estimates of the prevalence and dynamics of NSSI have been found to vary across countries and continents [3], it is possible that disparities in mediation findings may also reflect geographic and/or income level variations across study samples.

In sum, available mediation studies are characterized by marked heterogeneity in sampling and design, with inter-study variability serving to limit the confidence with which cross-study comparisons can meaningfully be made.

Comment 3: I identified two further studies in addition to Hu et al  Shenk et al that have specifically examined the relative mediating effects of depression end emotion dysregulation on the relationship between ACEs and NSSI. They are Wu et al Personality and individual differences, 2021-06, Vol.175, p.1107 and Gong et al Journal of interpersonal violence, 2024-03, Vol.39 (5-6), p.925-948

Response 3: Thank you for this comment. We have added Gong et al (Journal of Interpersonal Violence, 2024) but not Wu et al. 2021 (as this study did not consider child maltreatment as a risk factor for NSSI). Your comments encouraged us to conduct a further systematic search for further references that need to be added, with this search, combined with your suggestions, yielded at total of seven studies that were included in the paper.

  • Gong & Zhang, 2024
  • Hu et al., 2023
  • Peh et al., 2017
  • Shenk et al., 2010
  • Titelius et al., 2018
  • Wiguna et al., 2021
  • Wu et al., 2023

 Appropriate amendments were made in the text in relation to this extended list of studies:

              We were able to identify seven studies that have examined mediators of the ACEs-NSSI association [19,38-40,42-44]; with potential mediators considered in these studies being consistent with predictions suggested by contemporary conceptualizations of NSSI (i.e., ED, depression, and symptoms of posttraumatic stress disorder). Six studies considered ED as a potential mediator of the ACE-NSSI association, with ED being found to constitute a significant mediator or moderator of NSSI severity in 5 studies [38,39,42-44] but not in the study conducted by Shenk and associates [40]. The mediating effects of depression on NSSI severity were examined in  five studies, with depression being found to mediate NSSI outcomes in four studies, [19,38,39,42] but not in one study [40]. Finally, only one study [41] assessed the mediating effects of PTSD, with PTSD symptoms emerging as a significant mediator of NSSI severity, but with depression and ED not significantly mediating NSSI severity after controlling for PTSD.

Comment 4. If no hypotheses were entertained how was it the authors landed on ACEs as the risk factor and depressive symptoms, emotion dysregulation and posttraumatic stress symptoms as the mediating factors? Other putative risk factors for NSSI examined in research include:

Response 4: Thank you for this comment. While we acknowledge that there are many identified risk factors for NSSI, the risk factor (ACEs) and mediators (ED, depression and PTSD) considered in our study were informed by contemporary conceptualizations of risk for NSSI severity, as well as by available literature that has provided support (or partial support) for such conceptualizations. See for example:

Empirical support for both neurobiological and psychological pathways to NSSI has been reported in a number of studies. With respect to the neurobiological model, available studies have consistently reported a direct effect of ACES on NSSI [2,14,15,17-19,39,40] and a direct effect of ACEs on ED and/or emotional distress associated with PTSD and depression [2,41]. Similarly, with respect to the psychological model, studies have found that emotional distress and ED partially mediate the association between ACEs and NSSI [39,43].

Comment 5 . Stronger justification for choice of mediating variables is required, particularly as it does not align with the cited systematic review of Valencia-Aguda et al Journal of adolescence (London, England.), 2018-06, Vol.65 (1), p.25-38. Note also the long list of putative mediating variables reported in papers published since the systematic review.

Response 5: Thank you for these comments. We agree that the extant literature (Valencia-Agudo et al’s. review plus more recent references you provide) has identified a large number of putative variables for NSSI outcomes. However, we believe that we provide a compelling rationale for focusing on the mediators( ED, depression, and PTSD) that were included in the study.

  • Contemporary conceptualisations of NSSI outcomes strongly suggest that the selected variables are likely to serve as key mediating variables in the association between ACEs and NSSI outcomes:

psychological theories would predict that the association between ACEs and NSSI are likely to be mediated by feelings of distress and/or ED.

 A neurobiological perspective…would predict a direct effect of ACEs on the development of mental disorders (such as NSSI) as well as a direct effect of ACEs on (ED) and impaired executive functioning.

  • Available literature has provided compelling support that the selected variables can mediate the association between ACEs and NSSI outcomes:

Empirical support for both neurobiological and psychological pathways to NSSI has been reported in a number of studies. With respect to the neurobiological model, available studies have consistently reported a direct effect of ACES on NSSI [2,14,15,17-19,39,40] and a direct effect of ACEs on ED and/or emotional distress associated with PTSD and depression [2,41]. Similarly, with respect to the psychological model, studies have found that emotional distress and ED partially mediate the association between ACEs and NSSI [39,43].

In addition, we believe that it matters that the selected mediators are all readily modifiable, and thus amenable to therapeutic intervention.

Comment 6. How does the school population studied (predominantly black, majority male) influence the generalisability of the findings? What other confounders could be contributing to learner behavioural and psychological problems?

Response 6: Thank you for this comment. We agree that the study sample contained an over representation of males and Black African students, and that this might impact on the generalizability of study findings; with this being acknowledged as a limitation of the study:

Finally, male and Black African participants were over-represented in the study sample, which could potentially limit the generalizability of study findings. As such, further research, based on more demographically representative samples, is indicated.

Unfortunately, the survey did not collect information on other variables that could be considered as confounders of the ACEs-NSSI association.

Comment 7:  A novel aspect of the study was the inclusion of discrimination as an ACE item. The authors highlighted the importance of distinguishing types of ACE in NSSI research. I am puzzled therefore why different types of ACE was not included in the modelling?

Response 7: Thank you for this comment. While we do acknowledge that ACE measures need to assess a broad range for ACEs, we do not emphasize specific ACES  in our modelling as ACE measures are primary designed to assess the cumulative effect of ACEs (i.e., the number of different ACES experienced by an individual) Thus, for a 15-item ACE scale, scores can range from 0 (no ACEs) to 15 (all 15 ACEs experienced). We have attempted to make this explicit in the text:

Given that ACE questionnaires are primarily designed to measure the confluence of multiple types of ACEs [21], ACEs in this study were assessed using a cumulative measure of ACEs (i.e., the total number of ACEs reported by participants), with potential scores ranging from 0-15.

 Comment 8: The conclusions are generic and do not relate specifically to the study findings. With respect to secondary and tertiary prevention the authors need to place their recommendations ithin the context of what is currently known about treatment for NSSI.

Response 8: Thank you for these comments. We agree and have reorganised the Discussion section to more clearly align it with the study objectives and associated study findings. Although the originally submitted manuscript interventions for NSSI antecedents (ACEs and ED), we acknowledge that it did not directly address psychotherapeutic interventions for adolescents with NSSI. As such we have extended the discussion to incorporate interventions that have been found to be effective in addressing NSSI symptoms among adolescents.

Similarly, there are a number of psychotherapeutic interventions that have been found to be effective in treating NSSI among adolescents; with the most effective interventions, in terms of reducing NSSI behaviors, being interventions that use strategies from (or adapted from) Cognitive Behavioral Therapy, Dialectical Behavior Therapy and/or Psychoeducation [73].

Reviewer 2 Report

Comments and Suggestions for Authors

This study assessed risk factors for non-suicidal self-injury (NSSI) in a non-clinical sample of South African adolescents. Adverse childhood experiences (ACEs) being an identified risk factor for NSSI, its association with NSSI was explored as well as the role of potential mediators: emotion dysregulation, depressive symptoms and PTSD symptoms. The multi-mediation analysis revealed that emotion dysregulation partially mediated the association between ACEs and NSSI in this non-clinical sample of South African adolescents. One of the reason for which  this study was conducted is that findings from high-income countries may not generalize to low- to middle-income countries, such as South Africa. Implications for public health are discussed briefly. Strengths include an inclusive definition of ACES (including a measure of discrimination) and the usage of standardized measures with documented psychometric properties. Overall, this article is of significant scientific value. My main concern relates to the absence of confounders in the statistical analysis. Below are my general and detailed comments by sections.

-General comments

---Introduction

The introduction provides a solid background. The rationale for studying the factors and associations under consideration here is well presented, and empirically and theoretically grounded. The relevance of using an inclusive definition of ACEs, notably, the consideration of discrimination, is well argued. Overall, the rationale for this study is excellent, although its novel contribution could me more highlighted. The most important novel contribution of this study appears to be the examination of the associations between ACEs and NSSI, and potential mediators, in a low- to middle-income country. A few additions to the introduction could further highlight this point. 

---Methods

The methods are described adequately, and the design is appropriate. Strengths include the usage of standardized measures with documented psychometric properties. The description of the statistical analysis could be more precise : what variables were entered in the model and how were the models tested? It seems that all variables were entered simultaneously in the model, but precising this would clarify the methods and enhance transparency. 

Although socio-demographic characteristics of participants were documented, they do not appear to have been considered as confounders, despite that age was associated with NSSI (p. 5, l. 215). It would be helpful to understand whether confounders (especially age) were considered and, if not, why they were not included in the mediation model and how could this affect the interpreation of findings.

---Results and Discussion

The results are very clearly presented, and the conclusions appear to be supported by the results. The exclusive reliance on self-report measures could be identified as a limitation as well, and its implication for results could be discussed.

- Detailed comments

---Introduction

Regarding the studies by Hu and colleagues, and Shenk and colleagues, were they conducted in high-income or low-income countries? Adding this information could help highlight the novel contribution of this study (if they were conducted in high-income countries) or provide a basis for comparison. In addition, could the discordant findings be explained by the fact than Shenk and colleagues did consider PTSD symptoms?

---Methods

For the NSSI measure, please clarify the score used in analysis and the score range.

How many items does the subscale of the Patient Health Questionnaire (PHQ-9) comprise to assess the severity of depressive symptoms?

p.5, line 194: ‘In cases where distress was noted, participants were provided with an option of discontinuing their participation.’ Were resources for help provided?

Author Response

Manuscript: A multi mediation analysis of the association between adverse childhood experiences and non-suicidal self-injury among South African adolescents

RESPONSE TO REVIEWER 2 COMMENTS

Comment 1: The most important novel contribution of this study appears to be the examination of the associations between ACEs and NSSI, and potential mediators, in a low- to middle-income country. A few additions to the introduction could further highlight this point.

Response 1: Thank you for this comment. We agree and have addressed the issue you raise:

It is also important to note that estimates of the prevalence and dynamics of NSSI have been found to vary across countries and continents [3]. However, such variations have been reported almost exclusively in relation to countries located in the Global North; with two recent meta-analyses of NSSI prevalence rates [3,12] including only two studies that were conducted in the Global South (one in Brazil and one in Australia) but no studies that were conducted in the African context. As such, there would appear to be a clear need for global estimates of NSSI prevalence rates and dynamics to be expanded to reflect a more truly global perspective, with such expansion appearing to be most strongly indicated in relation to the African context.

The primary aim of this research was to explore the mediating role of emotion regulation, depression, and PTSD on the association between ACEs and NSSI in a non-clinical sample of South African adolescents, and to assess the extent to which study findings correspond to findings that have been reported in studies conducted largely in the Global North (mainly in Asia and North America).

Comment 2: The description of the statistical analysis could be more precise: what variables were entered in the model and how were the models tested? It seems that all variables were entered simultaneously in the model, but precising this would clarify the methods and enhance transparency.

Response 2: Thank you for this comment. We agree and have extended the description of the statistical analysis to encompass the issues you raise:

The data analysis involved three stages. Initially data were reviewed, with this review revealing that the proportion of missing values was less than 3% across all study variables, which can be considered negligible as it falls well below the 5% rule of thumb for acceptable levels of missing data [54]. As such, for each analysis, listwise deletion was used to handle missing values. The data review also revealed that there were no outliers.

Second, prevalence rates were calculated for ACEs and NSSI, with descriptive statistics (means, standard deviations and intercorrelations) being calculated for all key variables. Finally, a multi mediation analysis was conducted using the PROCESS macro for SPSS [55]. After controlling for the effects of demographic variables (age gender, and ethnicity): (a) the risk factor entered in this analysis was a cumulative measure of ACEs exposure (i.e., the number of different types of ACE reported by each participant); (b) the outcome variable being the severity of NSSI; and (c) the mediating variables being the severity of ED, depressive, and PTSD symptoms; and d) with bootstrapped confidence intervals (CI), based on 5000 bootstrapped samples, being used to determine the significance of indirect effects.

Comment 3: It would be helpful to understand whether confounders (especially age) were considered and, if not, why they were not included in the mediation model and how could this affect the interpretation of findings.

Response 3: Thank you for this comment. We agree. As indicated in our response to Comment 2 (above), demographic variables were entered in the mediation analysis as control variables – a practice that has been generally employed in the extant literature on mediators of the association between ACEs and NSSI.

Comment 4: The exclusive reliance on self-report measures could be identified as a limitation as well, and its implication for results could be discussed.

Response 4: Thank you for this comment. We agree, and have included the use of self-report measures as a study limitation:

In addition, the use of self-report measures of participants wellbeing falls short of the gold standard for assessing psychological wellbeing (i.e., a comprehensive clinical assessment). As such, future studies on the association between ACEs and NSSI would benefit from a reliance on a more comprehensive assessment of key constructs.

Comment 5: Regarding the studies by Hu and colleagues, and Shenk and colleagues, were they conducted in high-income or low-income countries? Adding this information could help highlight the novel contribution of this study (if they were conducted in high-income countries) or provide a basis for comparison. In addition, could the discordant findings be explained by the fact than Shenk and colleagues did consider PTSD symptoms?

Response 5: [Please note that following recommendations of another reviewer and an updated literature search by ourselves, a total of 7 mediation studies were identified].

We were able to identify seven studies that have examined mediators of the ACEs-NSSI association [19,38-40,42-44]; with potential mediators considered in these studies being consistent with predictions suggested by contemporary conceptualizations of NSSI (i.e., ED, depression, and symptoms of posttraumatic stress disorder

We agree with your comments and have provided an updated indication of possible reasons for inter-study disparities, that includes the issue of country income level you raise.

              Disparities in available studies may reflect differences in study design. With respect to sampling strategies, study participants were drawn from the general community in five studies [19,38,39,42,44], with the mediating effects on NSSI outcome being assessed in a clinical sample of adolescents in one study [43] and a sample drawn from Child Protection Services referrals in another study [40]. Further, there was a lack of consistency regarding the role played by key variables in relation to NSSI outcomes. Although ED and depression were generally conceptualized as constituting mediating variables, ED was entered as a moderating variable in one study [38], with depression being entered in another study as a control variable [42].

In interpreting available mediational findings it is also important to bear in mind that there were marked differences across studies in the way that NSSI was operationalised, with some studies employing validated research instruments in order to obtain  an estimate of NSSI severity (19,38,39,42,43], and two studies [40,44] relying exclusively on one or two broad questions to asses for NSSI (e.g., “Have you ever tried to intentionally hurt yourself by damaging the physical integrity of your body”). Clearly such broad questions fail to adequately address the nature and scope of NSSI as defined by the American Psychiatric Association [1].

At a broader level, available mediation studies have been conducted exclusively in two continents – North America [38,40,43] and Asia [19, 39,42,44] – and have been conducted among participants residing in middle income countries [19,39,44] or in high income-countries [38,40,42,43]; with none of the identified mediation studies having been conducted in countries located in the Global South. To the extent that estimates of the prevalence and dynamics of NSSI have been found to vary across countries and continents [3], it is possible that disparities in mediation findings may also reflect geographic and/or income level variations across study samples.

In sum, available mediation studies are characterized by marked heterogeneity in sampling and design, with inter-study variability serving to limit the confidence with which cross-study comparisons can meaningfully be made.

Comment 6: For the NSSI measure, please clarify the score used in analysis and the score range.

Response 6: Thank you for this comment. We agree and have added clarity regarding NSSI scoring and score ranges for the SH:

For each item on the SH subscale, participants were asked to indicate the number of days that they had engaged in such behavior in the past 12-months. Each item on this adapted version of the RTSHIA was scored using a 4-point scale (never to many times in the past year), providing a score range of 0-24.

Comment 7: How many items does the subscale of the Patient Health Questionnaire (PHQ-9) comprise to assess the severity of depressive symptoms?

Response 7: Thank you for this comment. We have clarified the number of items in the PHQ-9:

Symptoms of depression were assessed using a nine-item subscale of the Patient Health Questionnaire (PHQ-9) that is used to assess for the severity of depressive symptoms

Comment 8: In cases where distress was noted, participants were provided with an option of discontinuing their participation.’ Were resources for help provided?

Response 8: Thank you for this comment. We have clarified that support was offered to participants:

In addition, written gatekeeper permission from the school principal, parental consent, and participant assent were obtained; with continuous assent being employed to ensure that students could discontinue their participation at any time, and with offers of free supportive counselling from the school counsellor being made to all participants.

Round 2

Reviewer 1 Report

Comments and Suggestions for Authors

Thank you the authors have satisfactorily responded to the concerns I raised in my first review